

# An efficient Equilibrium Optimizer for parameters identification of photovoltaic modules

Essam H. Houssein[1], Gamela Nageh[1], Mohamed Abd Elaziz[2] and Eman Younis[1]

[1] Faculty of Computers and Information, Minia University, Minia, Egypt
[2] Department of Mathematics, Faculty of Science, Zagazig University, Zagazig, Egypt

## ABSTRACT

The use of solar photovoltaic systems (PVs) is increasing as a clean and affordable source of electric energy. The Pv cell is the main component of the PV system. To improve the performance, control, and evaluation of the PV system, it is necessary to provide accurate design and to define the intrinsic parameters of the solar cells. There are many methods for optimizing the parameters of the solar cells. The first class of methods is called the analytical methods that provide the model parameters using datasheet information or I–V curve data. The second class of methods is the optimization-based methods that define the problem as an optimization problem. The optimization problem objective is to minimize the error metrics and it is solved using metaheuristic optimization algorithms. The third class of methods is composed of a hybrid of both the analytical and the metaheuristic approaches, some parameters are computed by the analytical approach and the rest are found using metaheuristic optimization algorithms. Research in this area faces two challenges; (1) finding an optimal model for the parameters of the solar cells and (2) the lack of data about the photovoltaic cells. This paper proposes an optimization-based algorithm for accurately estimating the parameters of solar cells. It is using the Improved Equilibrium Optimizer algorithm (IEO). This algorithm is improved using the Opposition Based Learning (OBL) at the initialization phase of EO to improve its population diversity in the search space. Opposition-based Learning (OBL) is a new concept in machine learning inspired by the opposite relationship among entities. There are two common models for solar cells; the single diode model (SDM) and double diode model (DDM) have been used to demonstrate the capabilities of IEO in estimating the parameters of solar cells. The proposed methodology can find accurate solutions while reducing the computational cost. Compared to other existing techniques, the proposed algorithm yields less mean absolute error. The results were compared with seven optimization algorithms using data of different solar cells and PV panels. The experimental results revealed that IEO is superior to the most competitive algorithms in terms of the accuracy of the final solutions.

Corresponding author
Gamela Nageh,
gamela.nageh@mu.edu.eg

## INTRODUCTION

A major drawback of the use of fossil fuel is its negative environmental effects; therefore, alternative renewable energy sources (RESs) are used as an alternative clean and affordable energy source. There are seven types of RESs namely, biomass (*López et al., 2008*), geothermal (*Houssein, 2019*), wind (*Osama et al., 2017*), solar (*Hocaoglu, 2011*; *Munshi & Yasser, 2017*), hydroelectric (*Bagher et al., 2015*), hydrogen (*Dunn, 2002*), and tide (*Egbert & Ray, 2000*). Solar energy is considered one of the most important sources of renewable energy, as the sun plays a significant role in it's generation. A PV system uses solar cells (SCs) to convert the sunlight into electricity (*AbdElminaam, Said & Houssein, 2021*).

PV SCs are a type of RES that is clean (*i.e.*, generates no pollution), has a long life, has a simple design and easy to install. However, solar energy has several drawbacks compared with other RES. The main disadvantages of PV solar energy systems are the high initial exploitation and low capacity of SCs (*Navabi et al., 2015*). An open research topic is optimizing the efficiency of SCs. Because PV modules are installed outdoors, the weather conditions affect the generated energy, which results in a high maintenance cost and difficulty in control.

PV modules contain SCs, which consist of a semiconductor of monocrystalline material (KC200GT) (*Park et al., 2014*), monocrystalline (SM55) (*Salam, Ishaque & Taheri, 2010*), and thin film (ST40) (*Alam, Yousri & Eteiba, 2015*). The global settings of a SC depend on a circuit to examine the performance of the cell under various conditions, such as temperature and solar irradiance. A SC can be described by the following two popular models: the single-diode model (SDM) and the double-diode model (DDM) (*Lim et al., 2015*; *Humada et al., 2016*; *Ismaeel et al., 2021*). The SDM is similar to the DDM, and the variation in cellular performance is imperceptible. The SDM is more common than the DDM. The SDM model includes five unknown parameters, while the DDM has seven unknown parameters. Accurate determination of the unknown variables plays an important role in optimizing the SC and it's internal parameters.

The PV module manufacturer's data sheet contains the parameters of the open circuit voltage (Voc), voltage and current at the point of maximum power (Vmpp and Impp), short-circuit current (Isc), maximum power (Pmpp), voltage temperature and current temperature coefficients (Kv and Ki). These parameters can also be obtained experimentally. The SDM, however, requires other parameters, such as the reverse saturation current (Io), photogenerated current (Ipv), nonphysical ideality factor of the diode (a), shunt resistance (Rp), and series resistance (Rs). However, these parameters are not provided in the data sheet. Therefore, determining these parameters is a major research challenge.

There are three types of approaches for solving the parameter estimation problem: traditional methods (analytical methods) (*Easwarakhanthan et al., 1986*), metaheuristic methods (*AlRashidi et al., 2011*) and hybrid methods. Traditional methods use a modified nonlinear least squares estimation approach based on Newton's method. Traditional approaches depend on the starting point of the iterative method, which is a key problem.

As a result, these methods tend to be trapped in local solutions. An example of this type of techniques is presented in *Chan, Phillips & Phang (1986)*. Several features, such as convexity, continuity, and differentiability, are required for usability. The drawbacks of these methods include high demand of computational resources, sensitivity to initial conditions, and the tendency to provide sub-optimal solutions.

Unlike conventional methods, metaheuristic approaches have been proposed to increase the probability of obtaining the global solution in a reasonable time. In the literature, the problem of determining the parameters of the SC has been solved using genetic algorithms (GA) (*Ismail, Moghavvemi & Mahlia, 2013*), particle swarm optimization (PSO) (*Wei et al., 2011*), and simulated annealing (SA) (*El-Naggar et al., 2012*). Although these metaheuristic algorithms produce better solutions than conventional algorithms, they have some limitations (*Dai, Chen & Zhu, 2009*). The third approach for solving this problem is using hybrid methods which use a combination of the analytical and the metaheuristic approaches. A review of the methods of the parameter estimation of the SCs is presented in *Oliva et al. (2019)*.

Recently, researchers have used metaheuristic algorithms to extract the optimal parameters of PV systems (*Oliva et al., 2019*). These algorithms include the bird-mating optimizer (*Askarzadeh & Rezazadeh, 2013b*), artificial immune system (*Jacob et al., 2015*), repaired adaptive differential evolution (*Gong & Cai, 2013*), pattern search (*AlHajri et al., 2012*), harmony search-based algorithms (*Askarzadeh & Rezazadeh, 2012*), SA (*El-Naggar et al., 2012*), chaos particle swarm algorithm (*Wei et al., 2011*), artificial bee colony (*Askarzadeh & Rezazadeh, 2013a*) the mutative-scale parallel chaos optimization algorithm (*Yuan, Xiang & He, 2014*), adaptive differential evolution (*Chellaswamy & Ramesh, 2016*), gray wolf optimizer (GWO) (*Rodrguez & Murillo, 2017*), mine blast algorithm (*El-Fergany, 2015*), improved shuffled complex evolution algorithm (*Gao et al., 2018*), direct search optimization algorithm (*Osheba, Azazi & Shokralla, 2017*), evaporation-rate-based water cycle algorithm (*Gotmare et al., 2017*), Tabu search (*Siddiqui & Abido, 2013*), and chaos PSO (*Wei et al., 2011*).

An interesting approach to solve the problem of solar PV parameter estimation is based on the flower pollination algorithm (FPA) (*Alam, Yousri & Eteiba, 2015*). The FPA mimics flower pollination behavior. However, its results are inaccurate. The multi-verse optimizer has also been used to extract the parameters of the PV system. But, this algorithm also generates results far from the optimal solution (*Ali et al., 2016*). An optimization algorithm for artificial swarms of bees identifies parameters of the SC models and is inspired by the intelligent behaviors of the bees, such as collecting and processing nectar (*Askarzadeh & Rezazadeh, 2013a*). The simplified swarm optimization algorithm (*MSSO*) has been applied to both the SDM and DDM to minimize the square error between the experimental and calculated data. However, this minimization was not efficient $9.8607E^{-04}$ (*Lin et al., 2017a*). The improved chaotic whale optimization algorithm has also been used to estimate the parameters of the PV. However, it is complex and difficult to apply (*Oliva, El Aziz & Hassanien, 2017*). The most recent typologies in the literature are listed in Table 1.

**Table 1 Summary of the previous related research.**

| Year | Method | RMSE |
|------|--------|------|
| 2017 | MSSO (*Lin et al., 2017b*) | SDM: $9.8607E^{-04}$ |
|      |        | DDM: $9.8281E^{-04}$ % |
| 2017 | ABC with chaotic maps (*Oliva et al., 2017*) | SDM: $9.8602 \times 10^{-04}$ |
|      |        | DDM: $9.8262 \times 10^{-04}$ |
| 2017 | WDO (*Derick et al., 2017*) | SDM: 0.00084 |
|      |        | DDM: 0.00106 |
| 2017 | WOA with chaotic maps (*Oliva, El Aziz & Hassanien, 2017*) | SDM: $9.8602 \times 10^{-04}$ % |
|      |        | DDM: $9.8272 \times 10^{-04}$ |
| 2018 | SCE with a complex evolution strategy (*Gao et al., 2018*) | SDM: $9.8602 \times 10^{-04}$ |
|      |        | DDM: $9.8248 \times 10^{-04}$ |
| 2018 | WDO (*Mathew et al., 2017*) | DDM: 0.127 |
| 2018 | ImCSA (*Kang et al., 2018*) | SDM: $9.8602E^{-04}$ |
|      |        | DDM: $9.8249E^{-04}$ |
| 2019 | TLO (*Li et al., 2019*) | SDM: $9.8609E^{-04}$ |
|      |        | DDM: $9.8612E^{-04}$ |
| 2019 | ISCA (*Chen et al., 2019*) | SDM: $9.8602E^{-04}$ |
|      |        | DDM: $9.8237E^{-04}$ |
| 2020 | SMA (*Kumar et al., 2020*) | SDM: $9.8582E^{-04}$ |
|      |        | DDM: $9.8148E^{-04}$ |
| 2020 | COA (*Diab et al., 2020*) | SDM: $7.7547E^{-04}$ |
|      |        | DDM: $7.6480E^{-04}$ |

The above methods use metaheuristic algorithms that require large computational time and lead to high error rate between the experimental and the calculated data. Therefore, this work proposes an efficient algorithm called the improved equilibrium optimizer (IEO) algorithm to estimate the optimal parameters of the PV system. IEO was validated by comparing its results with those of a mathematical method and other approaches. Using the IEO, the minimum error between the estimated and the experimental data and the minimum convergence time for the optimal parameters and the best curve fit (I–V) was obtained. An accurate PV model based on IEO algorithm can play an important role in increasing the overall efficiency of a PV system. It guarantees the optimal use of the available solar energy, which results in reducing the costs of the PV system. To the best of our knowledge, the IEO is a new optimization algorithm whose potential has not yet been extensively applied in real problems. The motivation of this paper is to propose a parameter estimation technique for efficiently and accurately extracting the parameters of the SCs and PV modules. The major contributions of this work are as follows:

1. Proposing the Improved Equilibrium Optimizer (IEO) algorithm, which is inspired by the control, volume, mass, and balance models for estimating the photovoltaic (PV) parameters.

2. IEO algorithm proved that it can obtain the optimal value of the PV variables of single and double diode models.

3. Comparing the IEO with other metaheuristic algorithms the results show that IEO provided smaller errors between the calculated values and the actual values.

The remainder of this paper is organized as follows: 'Mathematical Model of Photovoltaic Cell/Module' offers a depiction of the mathematical model of the PV cell/module. Equilibrium optimizer (EO) and Improved Equilibrium Optimizer (IEO) are presented in 'Equilibrium Optimizer (EO) Algorithm' and 'Improved Equilibrium Optimizer (IEO) Algorithm', respectively. The modified objective function is introduced in 'Objective Function'. In 'Simulation Results and Analysis' and 'Results of Comparing the Effeciency of the Ieowith other Metaheuristic Algorithms' analysis and Comparative study of the proposed algorithm with other meta-heuristics algorithms are provided. Finally, the conclusions and future work are presented in 'Conclusions and Future Work'.

## MATHEMATICAL MODEL OF PHOTOVOLTAIC CELL/MODULE

This section presents the mathematical model of the PV unit (*Biswas et al., 2019*) that can be obtained by the SDM (*El-Arini, Othman & Fathy, 2013*) and DDM (*Chin et al., 2011*). Figure 1 displays the equipollent circuit diagram of a typical PV cell. The current ($I$) of this cell is as follows:

$$I = I_{PV,Cell} - I_D \tag{1}$$

where, $I_{PV,Cell}$ is the PV current and $I_D$ is the diode current.

$$I_D = I_{O,Cell} \left[ exp\left(\frac{qV}{aKT}\right) - 1 \right] \tag{2}$$

where, $I_{O,Cell}$ is the reverse saturation current of the diode, $q$ ($1.60217646 \times 10^{-19}$ ($C$)) is the charge of an electron, $k$ ($1.38064852 \times 10^{-23}$ $J/K$) is the Boltzmann constant, $T$ is the temperature (in Kelvin) of the diode junction, and $a$ is the ideal factor of the diode. Therefore, Eq. (1) can be rewritten to obtain the I–V characteristic of a typical cell as follows:

$$I = I_{PV,Cell} - I_{O,Cell} \left[ exp\left(\frac{qV}{aKT}\right) - 1 \right] \tag{3}$$

PV cells can be linked either in parallel or in series to form a PV system. Here, $N_c$ represents the number of cells that are linked in series.

### Single diode model

In the SDM, five unknown parameters must be obtained. These parameters are the photon current ($I_{pv}$), reverse saturation current of the diode ($I_o$), the ideality factor ($a$), and the series and parallel resistances ($R_s$ and $R_p$). Figure 2 presents the SDM of a PV cell with $N_C$ series-connected cells.

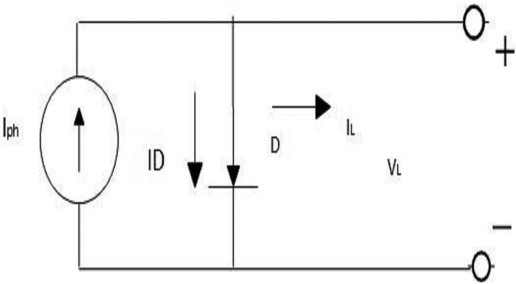

**Figure 1 Equivalent circuit of an idealphotovoltaic (PV) cell.**

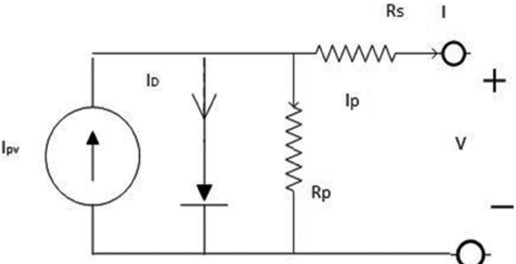

**Figure 2 Single-diode model ofphotovoltaic (PV) cell.**

Therefore, the $I-V$ characteristics of the SDM of the PV are represented as:

$$I = I_{PV} - I_O \left[ exp \frac{q(V + R_S I)}{aKN_C T} - 1 \right] - \frac{V + R_S I}{R_S} \tag{4}$$

At the point of the open circuit, we find that $(V = V_{OC})$ and $(I = 0)$ (opinion); thereafter, from Eq. (4), we obtain;

$$0 = I_{PV} - I_O \left[ exp \left( \frac{qV_{OC}}{aKN_C T} \right) - 1 \right] - \frac{V_{OC}}{R_P} \tag{5}$$

Thus, $\quad I_{PV} = I_O \left[ exp \left( \frac{qV_{OC}}{aKN_C T} \right) - 1 \right] - \frac{V_{OC}}{R_P} \tag{6}$

At the point of the short circuit $(I = I_{SC})$ and $(V = 0)$ (opinion); thereafter, from Eq. (4) we obtain;

$$I_{SC} = I_{PV} - I_O \left[ exp \left( \frac{qR_S I_{SC}}{aKN_C T} \right) - 1 \right] - \frac{R_S I_{SC}}{R_P} \tag{7}$$

*Therefore,* $\quad I_{PV} = I_{SC} + I_O \left[ exp \left( \frac{qR_S I_{SC}}{aKN_C T} \right) - 1 \right] - \frac{R_S I_{SC}}{R_P} \tag{8}$

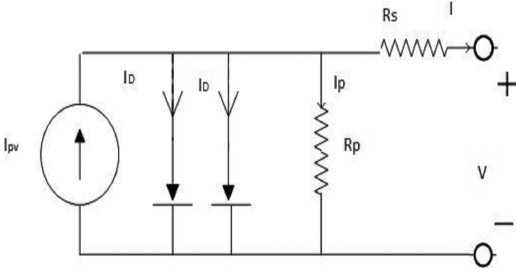

**Figure 3  Double diode model of the PVcell.**     

From Eqs. (6) and (8) we have;

$$I_O = \frac{I_{SC} + \dfrac{R_S I_{SC}}{R_P} - \dfrac{V_{OC}}{R_P}}{exp\left(\dfrac{qV_{OC}}{aKN_C T}\right) - exp\left(\dfrac{qR_S I_{SC}}{aKN_C T}\right)} \tag{9}$$

Substituting Eq. (9) into Eq. (6) yields;

$$I_{PV} = \frac{\left(I_{SC} + \dfrac{R_s I_{SC}}{R_P} - \dfrac{V_{OC}}{R_P}\right)\left[exp\left(\dfrac{qV_{OC}}{aKN_C T} - 1\right)\right]}{exp\left(\dfrac{qV_{OC}}{aKN_C T}\right) - exp\left(\dfrac{qR_S I_{SC}}{aKN_C T}\right)} \tag{10}$$

At the maximum power point ($V = V_{MPP}$) and ($I = I_{MPP}$) (say), from Eq. (4) we have;

$$I_{MPP} = I_{PV} - I_O \left[exp\frac{q(V_{MPP} + R_S I_{MPP})}{aKN_C T} - 1\right] - \frac{V_{MPP} + R_s I_{MPP}}{R_P} \tag{11}$$

## Double-diode model

In the DDM in Fig. 3, seven unknown parameters must be obtained. These parameters are the photon current ($I_{pv}$), reverse saturation current of the first diode ($I_{O1}$), reverse saturation current of second diode ($I_{O2}$), ideality factors of the first and second diodes ($a_1$) and $a_2$, and the PV cell series and parallel resistances ($R_S$ and $R_P$). Therefore, the I-V characteristics of the DDM are as follows:

$$I = I_{PV} - I_{D1} - I_{D2} - \frac{V + R_s I}{R_P} \tag{12}$$

i.e.,     $$I = I_{PV} - I_{O1} \left[exp\frac{q(V + R_S I)}{a_1 KN_C T} - 1\right] - I_{O2}\left[exp\frac{q(V + R_S I)}{a_2 KN_C T} - 1\right] - \frac{V + R_S I}{R_P} \tag{13}$$

At the point of the open circuit, $I = 0$ and $V = V_{OC}$; thus, from Eq. (13) we obtain;

$$0 = I_{PV} - I_{O1} \left[exp\frac{q(V + R_S I)}{a_1 KN_C T} - 1\right] - I_{O2}\left[exp\frac{q(V + R_S I)}{a_2 KN_C T} - 1\right] - \frac{V_{OC}}{R_S} \tag{14}$$

Therefore,
$$I_{PV} = O1\left[exp\frac{q(V+R_S I)}{a_1 K N_C T} - 1\right] - I_{O2}\left[exp\frac{q(V+R_S I)}{a_2 K N_C T} - 1\right] - \frac{V_{OC}}{R_S} \quad (15)$$

At the point of the short circuit, $V = 0$ and $I = I_{SC}$; thus, from Eq. (13) we obtain;

$$I_{SC} = I_{PV} - I_{O1}\left[exp\left(\frac{qR_S I_{SC}}{a_1 k N_C T}\right) - 1\right] - I_{O2}\left[exp\left(\frac{qR_S I_{SC}}{a_2 k N_C T}\right) - 1\right] - \frac{R_S I_{SC}}{R_P} \quad (16)$$

Therefore,
$$I_{PV} = I - SC + I_{O1}\left[exp\left(\frac{qR_S I_{SC}}{a_1 k N_C T}\right) - 1\right]$$
$$+ I_{O2}\left[exp\left(\frac{qR_S I_{SC}}{a_2 k N_C T}\right) - 1\right] + \frac{R_S I_{SC}}{R_P} \quad (17)$$

From Eqs. (15) and (17) we have;

$$I_{O2} = \frac{I_{SC} + \dfrac{R_S I_{SC}}{R_P} - \dfrac{V_{OC}}{R_P} - I_{O1}\left[exp\left(\dfrac{qV_{OC}}{a_1 k N_C T}\right) - exp\left(\dfrac{qR_S I_{SC}}{a_1 k N_C T}\right)\right]}{exp\left(\dfrac{qV_{OC}}{a_2 k N_C T}\right) - exp\left(\dfrac{qR_S I_{SC}}{a_2 k N_C T}\right)} \quad (18)$$

Substituting Eq. (18) into Eq. (15) yields;

$$I_{PV} = I_{O1}\left[exp\left(\frac{qR_S I_{SC}}{a_1 k N_C T}\right) - 1\right]$$
$$+ \frac{I_{SC} + \dfrac{R_S I_{SC}}{R_P} - \dfrac{V_{OC}}{R_P} - I_{O1}\left[exp\left(\dfrac{qV_{OC}}{a_1 k N_C T}\right) - exp\left(\dfrac{qR_S I_{SC}}{a_1 k N_C T}\right)\right]}{\left[exp\left(\dfrac{qV_{OC}}{a_2 k N_C T}\right) - exp\left(\dfrac{qR_S I_{SC}}{a_2 k N_C T}\right)\right]\left[exp\left(\dfrac{qV_{OC}}{a_2 k N_C T}\right) - 1\right]^{-1}} + \frac{V_{OC}}{R_P} \quad (19)$$

At the maximum power point, $V = V_{MPP}$ and $I = I_{MPP}$ (say); then, from Eq. (13) we obtain;

$$I_{MPP} = I_{PV} - I_{O1}\left[exp\frac{q(V_{MPP}+R_S I_{MPP})}{a_1 k N_C T} - 1\right] - I_{O2}\left[exp\frac{q(V_{MPP}+R_S I_{MPP})}{a_2 k N_C T} - 1\right]$$
$$- \frac{V_{MPP}+R_S I_{MPP}}{R_P} \quad (20)$$

## EQUILIBRIUM OPTIMIZER (EO) ALGORITHM

In this section, we discuss the exploratory and exploitative phases of EO, inspired by the general mass balance equation for a control volume (*Faramarzi et al., 2020*).

### Initialization and evaluation of functions

Like many metaheuristic algorithms, the EO uses the initial population for beginning the optimization process. The processing in this method is divided into two phases: the first phase is the "teacher phase," which means learning from the teacher, and the second phase is the "learner phase," which means learning by interacting with other learners.

These phases are consequently iterated to produce better results until convergence is achieved. In EO, each particle (solution) with its concentration (position) acts as a search agent. Some solutions and their dimensions experience mutation, defined by a function, and selected by a parameter such as mutation probability and percentage The initial concentration is structured according to the particle number and dimensions, with normal initialization appearing as follows:

$$C_i^{initial} = C_{min} + rand_i(C_{max} - C_{min}) \qquad i = 1, 2, \ldots, n \tag{21}$$

$C_i^{initial}$ is the initial condensation of the ith particle, $C_{max}$ and $C_{min}$ are the maximum and minimum of the dimensions, respectively, and $rand_i$ is in the domain [0, 1] so that $n$ is the particle number representing a population. Particles are examined and thereafter classified as candidates.

### Equilibrium pool and candidates $C_{eq}$

The equilibrium case takes into consideration the last case of convergence of the algorithm, which should be the global optimum. At the beginning of the optimization algorithm, the equilibrium state is unknown, and it is only determined by the candidate that the equilibrium data creates a research model for particles. Depending on various factors, these candidates are identified, four particles specific to the optimization, and the condensation of others particle is the arithmetic mean of those. The number of candidates is random and depends on the nature of the problem. For example, the GWO uses the three best candidates ($\beta$, $\gamma$ and $\alpha$ wolves) to change place with other wolves. More than four candidates can positively impact these five particles and are used as candidates for balancing to build a vector that is the equilibrium set in Eq. (22):

$$\vec{C}_{eq,pool} = \{\vec{C}_{eq(1)}, \vec{C}_{eq(2)}, \vec{C}_{eq(3)}, \vec{C}_{eq(ave)}\} \tag{22}$$

Every particle at every repetition changes its condensation to select randomly from the candidate's selection with the same prospect.

### Exponential term ($F$)

The following exponential expression ($F$) contributes to the rule of changing condensation. A precise introduction of this expression leads to the balance of exploration and exploitation. The fluctuation range can change over time in an actual dominance volume, and should be a vector in the domain of [0, 1] as in Eq. (23):

$$\vec{F} = e^{-\vec{\lambda}(t-t_o)} \tag{23}$$

Thus, time, $t$, known as the task of iteration (*Iter*), reduces the repetition number of Eq. (24):

$$t = \left(1 - \frac{Iter}{Max\_iter}\right)^{\left(a_2 \frac{Iter}{Max\_iter}\right)} \tag{24}$$

where, *Max_iter* and *Iter* represent the maximum number of iterations and the current,

respectively, and a2 is a constant utilized to administer the capacity of use. To warranty concourse and improve the exploration and exploitation capacity of the EO, this is given as;

$$\vec{t_o} = \frac{1}{\lambda}\ln(-a_1 sign(\vec{r} - 0.5)[1 - e^{-\vec{\lambda}t}]) + t \tag{25}$$

Here, $a_1$ is a constant that controls the exploration capacity, while $a_2$ is a constant that controls the exploitation ability. Furthermore, r is a vector in the range [0–1] selected randomly. $Sign(r - 0.5)$ impacts the direction of exploration and exploitation. $a_1$ and $a_2$ are two and one, respectively, for all problems. The derived form of Eq. (23) with the substitution Eq. (25) is represented in Eq. (26) as follows:

$$\vec{F} = a_1 sign(\vec{r} - 0.5)[e^{-\vec{\lambda}t} - 1] \tag{26}$$

## Generation rate (G)

$G$ is another key expression of the proposed technique for providing the precise solution by enhancing exploitation. For example, a multipurpose model that characterizes $G$ as the top-order exponential decomposition operation is as follows:

$$\vec{G} = \vec{G_o}e^{-\vec{K}(t-t_o)} \tag{27}$$

where, $G_o$ is the initial value and $k$ represents the attenuation constant. For restricting the random variables, suppose that $K = \lambda$, and utilize the formerly derived exponential expression. The conclusive set of $G$ is as follows:

$$\vec{G} = \vec{G_o}e^{-\vec{\lambda}(t-t_o)} = \vec{G_o}\vec{F} \tag{28}$$

$$where: \quad \vec{G_o} = \vec{G}CP(\vec{C_{eq}} - \vec{\lambda}\vec{C}) \tag{29}$$

$$\vec{G}CP = \begin{cases} 0.5r_1 & r_2 < GP \\ 0 & r_2 \geq GP \end{cases} \tag{30}$$

Here, $GCP$ is the control parameter for the generation rate that contains the potential of the generation expression assistance for the update procedure. In addition, $r_1$ and $r_2$ are random vectors in [0, 1]. Finally, the EO update rule is as in Eq. (31):

$$\vec{C} = \vec{C_{eq}} + (\vec{C} - \vec{C_{eq}}).\vec{F} + \frac{\vec{G}}{\vec{\lambda}V}(1 - \vec{F}) \tag{31}$$

where, $F$ is presented in Eq. (26), and $V$ is a unit. To summarize all the aforementioned steps, a framework for EO is drawn in Fig. 4. It displays a conceptual sketch of the cooperation of all candidates for balance in a set of sample particles and how they sequentially affect the concentration update in the proposed algorithm. Since the topological positions of the equilibrium candidates are different in the initial iteration, and the exponential term generates large random numbers, this step-by-step update process helps particles that cover the entire area of the research. In the latter case, the opposite scenario occurs iterations when candidates circle the sweet stop with similar attitudes.

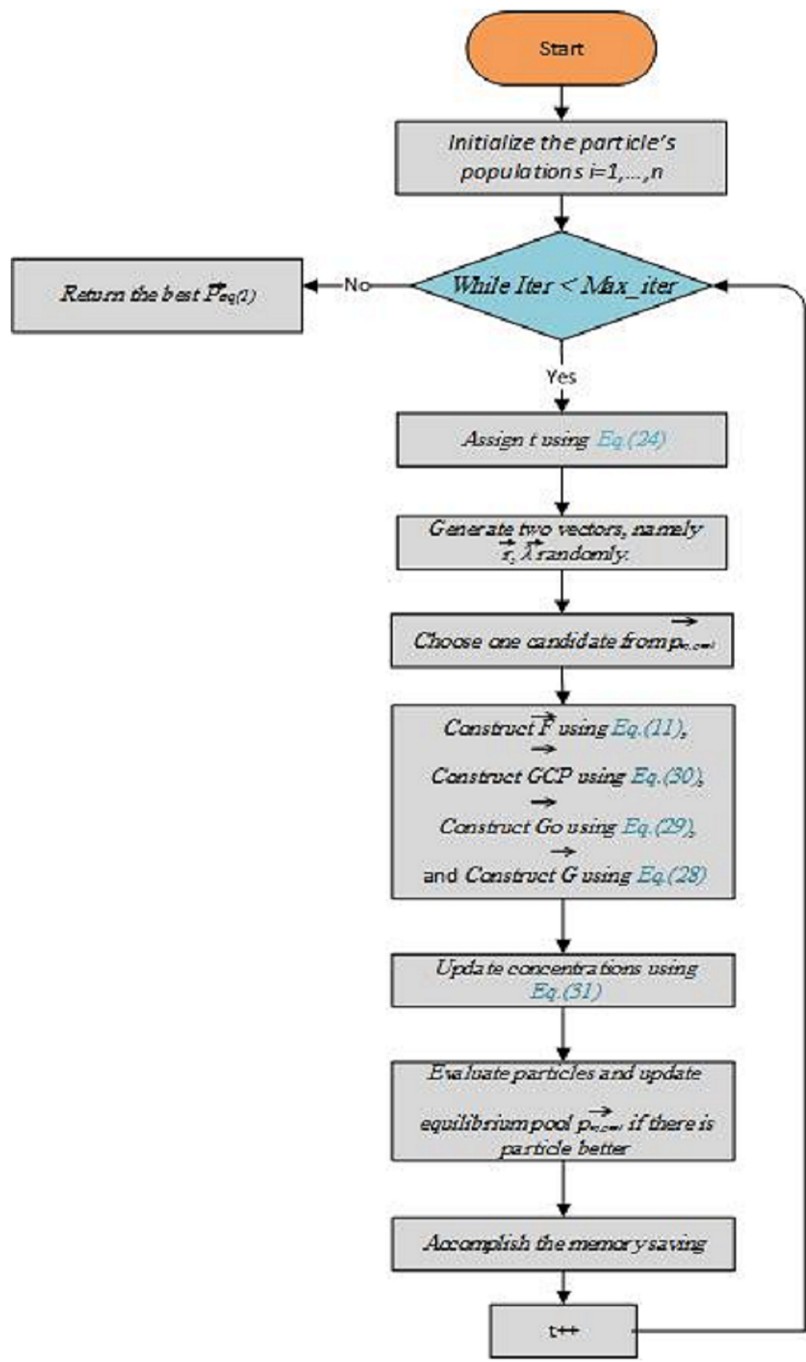

**Figure 4 A flowchart of the equilibrium optimizer (EO) algorithm.**

To this sometimes the exponential term generates small random numbers, which helps in refining the solutions. Providing smaller step sizes. This concept can also be extended to higher dimensions such as hyperspace in which the concentration is updated with the movement of the particle in $n$-dimensional space.

The EO algorithm has been tested for three known engineering design problems. A simple method of constraint management, the static penalty, is used here to punish the objective function with high score when constraints are violated at any level from their defined limits. The penalty coefficient must be large enough to correctly punish the objective function under the conditions of equality/inequality. Note that all technical test problems ran with the same number of iterations (500) and particles (30) according to the math functions described above.

## IMPROVED EQUILIBRIUM OPTIMIZER (IEO) ALGORITHM

We improved the Equilibrium Optimizer algorithm to give the best result that is closest to the global optimum. The improvement has been done by using opposition-based learning (OBL). The Equilibrium Optimizer algorithm with OBL gave better values to the unknown parameters of PV than Equilibrium optimizer without improvement.

### Opposition based learning (OBL)

The basic concept of Opposition-Based Learning (OBL) was originally introduced by *Tizhoosh (2005)*. The basic aim of this algorithm is optimization, to find the best candidate solution, the simultaneous consideration of an estimate and its corresponding opposite estimate which is closer to the global optimum. This has been achieved in a very short time and it has been used in various areas of soft computing. Opposition-based Learning (OBL), which has been used for improving convergence of the metaheuristic techniques for finding the global solution of the optimization problem. OBL theory has been applied to solve many real-world problems such as optimization and estimation of photovoltaic parameters. In general, any metaheuristic algorithm begins with creating an initial population as an attempt for finding the optimum solution(s), these initial solutions are generated randomly or based on a specific search range.

### *Opposite number*

x is defined as a real number over the domain $x \in [lb, ub]$. The opposite of $x$ is represented by $\bar{x}$ and to identified as:

$$\bar{x} = ub + lb - x \tag{32}$$

where ub and lb are the upper and lower bound of the search space. Eq. (32) can be generalized to be used in a search area with multi-dimensions. Therefore, to generalize it, every search-agent position and its opposite position will be introduced as the following:
x = $[x_1, x_2, x_3, x_4, \ldots, x_D]$ and $\bar{x} = [\bar{x}_1, \bar{x}_2, \bar{x}_3, \bar{x}_4, \ldots, \bar{x}_D]$ The values of all parameters in $\bar{x}$ will be estimated as:

$$\bar{x}_j = ub_j + lb_j - x_j \qquad \text{where } j = 1, 2, 3, \ldots, D \tag{33}$$

### *Opposition based learning*

In this optimization strategy if the fitness function $f(x)$ of its original solution, then $x = \bar{x}$; otherwise $x = x$.

**Table 2 Optimized parameters for the single-diode and double-diode model.**

| Models | Optimized variables | Calculated variables |
|---|---|---|
| Single-diode model | a, Rs, Rp | Io (Eq. (9)) and Ipv (Eq. (10)) |
| Double-diode model | a1, a2, Rs, Rp, Io1 | Io2 (Eq. (18)) and Ipv (Eq. (19)) |

## OBJECTIVE FUNCTION

The major purpose of estimating the parameters for SDM and DDM is to detect the parameter values that reduce the differences between the measured and calculated current that can be determined with an objective function. Table 2 presents details about the parameters that are optimized by IEO technique. The parameters were computed depending on the special connections that were discussed in 'Mathematical Model of Photovoltaic Cell/Module' for the SDM and DDM. For the DDM, $(a_1), (a_2), (R_S), (R_P)$ and $(I_{O1})$ are optimized $(I_{pv})$ and $(I_{o2})$ are computed by variables that are optimized utilizing the relationship discussed in 'Mathematical Model of Photovoltaic Cell/Module'. Metaheuristic algorithms are necessary to minimize errors at three points (open circuit $[0, V_{OC}]$, short circuit $[I_{SC}, 0]$, and points of maximum power $[I_{MPP}, V_{MPP}]$). For the SDM, the error at the open circuit (from Eq. (6)) is;

$$err_{OC} = I_O \left[ exp\left( \frac{qV_{OC}}{akN_CT} \right) - 1 \right] + \frac{V_{OC}}{R_P} - I_{PV} \tag{34}$$

The error at the short circuit (from Eq. (8)) is;

$$err_{SC} = I_{SC} + I_O \left[ exp\left( \frac{qR_SI_{SC}}{akN_CT} \right) - 1 \right] + \frac{R_SI_{SC}}{R_P} - I_{PV} \tag{35}$$

The error at the maximum power point (from Eq. (11)) is;

$$err_{MPP} = I_{PV} - I_O \left[ exp\left( \frac{q(V_{MPP} + R_SI_{MPP})}{akN_CT} \right) - 1 \right] - \frac{V_{MPP} + R_SI_{MPP}}{R_P} - I_{MPP} \tag{36}$$

The same applies for the error values of the DDM, which can be derived using Eqs. (15), (17) and (20) at these three points. The goal of optimization is considered the sum of the quadratic errors:

$$ERR = err_{OC}^2 + err_{SC}^2 + err_{MPP}^2 \tag{37}$$

## SIMULATION RESULTS AND ANALYSIS

To verify the accuracy, reliability, and robustness of the IEO algorithm, we evaluate the performance of the IEO algorithm for estimating the parameters of a PV cell, namely, the SDM and DDM. For every SDM and DDM for all SC types (polycrystalline, thin-film, and monocrystalline), the IEO were run separately 30 times. A MATLAB model was developed for testing the algorithms presented above. In addition, variables such as open-circuit voltage $(V_{OC})$, short circuit current $(I_{SC})$, maximum power point voltage

**Table 3 Electrical variables for photovoltaic (PV) cell under the standard test conditions.**

| Variables | Kyocera KC200G | Shell SQ85 | Shell ST40 |
|---|---|---|---|
| Kind | Polycrystalline | Monocrystalline | Thin film |
| Open circuit voltage, $V_{oc}$ (volt) | 32.9 | 22.2 | 23.3 |
| Short circuit current, $I_{sc}$ (amp) | 8.21 | 5.45 | 2.68 |
| Voltage at maximum power, $V_{MPP}$ (volt) | 26.3 | 16.6 | 17.2 |
| Current at maximum power, $I_{MPP}$ (amp) | 7.61 | 4.95 | 2.41 |
| Temperature coefficient of $V_{oc}$, $K_{V,OC}$ | −0.123 | −0.0725 | −0.1 (volt/°C) |
| Temperature coefficient of $I_{SC}$, $K_{I,SC}$ | $3.18 \times 10^{-10}$ | $0.8 \times 10^{-10}$ | $0.35 \times 10^{-10}$ (volt/°C) |
| Number of cells in series, $N_C$ | 54 | 36 | 36 |

**Table 4 Input variables to the Improved Equilibrium Optimizer (IEO) algorithm.**

| Parameters | SDM | DDM |
|---|---|---|
| Dimension of problem $d$ | 3 | 5 |
| Initial population size, $N_{pini}$ | 50 | 80 |
| Maximum no. of fitness evaluation, $NFE_{max}$ | 50 | 60 |
| The range of parameters, $[X_{min}, X_{max}]$ | $a = [0.1, 2.0]$ | $a_1 = [0.1, 2.0]$ |
| | | $a_2 = [0.1, 2.0]$ |
| | $R_s = [0.0001, 1]$ ohm | $I_{o1} = [10^{-12}, 10^{-06}]$ amp |
| | $R_p = [40; 200]$ ohm | $R_s = [0.0001, 1]$ ohm |
| | | $R_p = [40, 200]$ ohm |

($V_{MP}$), maximum power point current ($I_{MP}$), voltage temperature coefficient ($K_V$), and current temperature coefficient ($K_i$) were provided in the manufacturer's data sheet. All characteristics passed through three specified points are illustrated in Table 3 as follows: Open circuit: $V = V_{OC} = 32.9$ volt, $I = 0$ amp; Short circuit: $V = 0$ volt, $I = I_{SC} = 8.21$ amp; Maximum power: $V = V_{MPP} = 26.3$ volt, $I = I_{MPP} = 7.61$ amp. The algorithm was run on the MATLAB R2010a platform on a system with 4 GB RAM and an i3 processor. Improved code of equilibrium optimization using opposition-based learning is available in supplemental Source_Code file. Code to generated datasets used to test the code is available inside supplemental Datasets File. After improved Equilibrium optimization by opposition-based learning, we found the algorithm found best error than before.

## Numerical data

This study examined three types of SCs: polycrystalline, thin-film, and monocrystalline. Exemplary data supplied by the industries are displayed in Table 3. This table illustrates which current and voltage data at the three main points are explicitly specified at standard test conditions, such as irradiance of 1,000 w/m2 and temperature of 25 °C. Additional details of the parameters measured by the aforementioned equations are listed in Table 4.

**Table 5 Optimal variables estimated by the IEO for the single-diode model of polycrystalline, Kyocera KC200GT.**

| Run-No | Optimized parameters | | | Calculated parameters | | Error |
|---|---|---|---|---|---|---|
| | $a$ | $R_S$ (ohm) | $R_P$ (ohm) | $I_O$ (amp) | $I_{PV}$ (amp) | $E$ |
| 1 | 0.4105 | 0.0007 | 43.839 | 3.07E−27 | 8.2101 | 0 |
| 2 | 0.3494 | 0.001 | 43.835 | 5.06E−32 | 8.2102 | 0 |
| 3 | 1.0464 | 0.001 | 50.5571 | 1.37E−10 | 8.2102 | $1.39E-28$ |
| 4 | 1.1205 | 0.001 | 53.4895 | 7.05E−10 | 8.2102 | $9.66E-26$ |
| 5 | 0.3712 | 0.0008 | 43.8341 | 3.91E−30 | 8.2102 | 0 |
| 6 | 1.3315 | 0.0449 | 78.0334 | 2.81E−08 | 8.2147 | $6.53E-27$ |
| 7 | 1.6246 | 0.001 | 151.2808 | 9.62E−07 | 8.2101 | 0 |
| 8 | 0.4774 | 0.001 | 43.8344 | 2.13E−23 | 8.2102 | 0 |
| 9 | 1.3325 | 0.0089 | 70.3248 | 2.84E−08 | 8.211 | $3.94E-30$ |
| 10 | 0.3517 | 0.2152 | 43.6181 | 8.21E−32 | 8.2505 | 0 |
| 11 | 1.0185 | 0.0002 | 49.6954 | 6.93E−11 | 8.21 | 0 |
| 12 | 0.416 | 0.0753 | 43.7651 | 7.13E−27 | 8.2241 | 0 |
| 13 | 0.9492 | 0.001 | 43.8343 | 1.07E−11 | 8.2102 | 0 |
| 14 | 1.2509 | 0.1403 | 95.3509 | 8.12E−09 | 8.2221 | $7.59E-25$ |
| 15 | 0.5229 | 0.0104 | 43.8229 | 2.37E−21 | 8.2119 | 0 |
| 16 | 1.447 | 0.1103 | 181.9726 | 1.37E−07 | 8.2149 | 0 |
| 17 | 0.4153 | 0.0374 | 43.7961 | 6.38E−27 | 8.217 | 0 |
| 18 | 0.4778 | 0.0004 | 43.8338 | 2.22E−23 | 8.2101 | $3.16E-30$ |
| 19 | 0.9559 | 0.0007 | 43.8341 | 1.30E−11 | 8.2101 | $1.58E-25$ |
| 20 | 0.7702 | 0.001 | 44.991 | 1.89E−14 | 8.2102 | $7.61E-28$ |
| 21 | 1.6672 | 0.0009 | 190.3722 | 1.45E−06 | 8.21 | 0 |
| 22 | 1.6277 | 0.0009 | 153.5346 | 9.93E−07 | 8.2101 | $2.84E-29$ |
| 23 | 1.675 | 0.001 | 200 | 1.56E−06 | 8.21 | $2.23E-25$ |
| 24 | 1.1749 | 0.2297 | 135.4552 | 2.16E−09 | 8.2239 | 0 |
| 25 | 0.9312 | 0.0498 | 43.7842 | 6.33E−12 | 8.2193 | $7.88E-31$ |
| 26 | 0.8439 | 0.0009 | 45.863 | 3.58E−13 | 8.2102 | 0 |
| 27 | 0.9436 | 0.0013 | 47.7275 | 9.17E−12 | 8.2102 | 0 |
| 28 | 1.6677 | 0.0008 | 193.4348 | 1.46E−06 | 8.21 | 0 |
| 29 | 0.4168 | 0.1344 | 43.7342 | 7.98E−27 | 8.2352 | 0 |
| 30 | 1.266 | 0.0709 | 74.3245 | 1.03E−08 | 8.2178 | 0 |

## Results of the single diode model

Every group of the optimal variables was estimated by several runs led to zero error at the three main I–V points. There were three types of SCs: Polycrystalline cell Kyocera KC200GT, monocrystalline cell Shell SQ85, and thin film ST40. The run of each type is presented in this subsection. Table 5 presents the optimal parameters of the SDM for the polycrystalline cell Kyocera KC200GT using the IEO. Table 5 also presents the accuracy of the IEO in generating the PV parameters. The results of the best run of the

**Table 6 Optimal variables estimated by the IEO for the single-diode model of monocrystalline photovoltaic cell, Shell SQ85.**

| Run-No | Optimized parameters | | | Calculated parameters | | Error |
|---|---|---|---|---|---|---|
| | $a$ | $R_S$ (ohm) | $R_P$ (ohm) | $I_O$ (amp) | $I_{PV}$ (amp) | $E$ |
| 1 | 0.4952 | 0.001 | 42.4047 | 5.19E−23 | 5.1979 | 0 |
| 2 | 1.9999 | 0.0248 | 142.2184 | 1.08E−05 | 5.3762 | 0 |
| 3 | 1.9975 | 0.0478 | 179.6755 | 1.07E−05 | 5.3926 | 0 |
| 4 | 2 | 0.0381 | 162.6624 | 1.09E−05 | 5.3861 | 0 |
| 5 | 2.0000 0 | 0.0546 | 200 | 1.09E−05 | 5.3987 | 0 |
| 6 | 2 | 0.0009 | 115.7362 | 1.08E−05 | 5.3576 | $67.88E−31$ |
| 7 | 1.1357 | 0.0031 | 45.5716 | 4.76E−10 | 5.2158 | 0 |
| 8 | 1.9955 | 0.001 | 113.9509 | 1.04E−05 | 5.3562 | 0 |
| 9 | 2 | 0.001 | 115.7355 | 1.07E−05 | 5.3576 | 0 |
| 10 | 0.5286 | 0.151 | 41.3043 | 1.48E−21 | 5.2209 | 0 |
| 11 | 1.9986 | 0.001 | 115.1735 | 1.07E−05 | 5.3572 | $1.55E−28$ |
| 12 | 0.322 | 0.001 | 42.401 | 2.26E−35 | 5.1978 | 0 |
| 13 | 2 | 0.001 | 115.7355 | 1.08E−05 | 5.3576 | 0 |
| 14 | 2 | 0.0009 | 115.7381 | 1.08E−05 | 5.3576 | 0 |
| 15 | 1.6533 | 0.2203 | 200 | 7.02E−07 | 5.4055 | 0 |
| 16 | 2 | 0.0009 | 115.7466 | 1.08E−05 | 5.3576 | 0 |
| 17 | 0.2549 | 0.1756 | 40.2245 | 1.12E−44 | 5.2198 | 0 |
| 18 | 2 | 0.0546 | 200 | 1.09E−05 | 5.3987 | 0 |
| 19 | 2 | 0.001 | 115.7355 | 1.08E−05 | 5.3576 | 0 |
| 20 | 1.9986 | 0.001 | 115.1901 | 1.07E−05 | 5.3572 | 0 |
| 21 | 1.9999 | 0.0546 | 199.8438 | 1.09E−05 | 5.3987 | 0 |
| 22 | 0.3164 | 0.0514 | 41.3486 | 5.44E−36 | 5.2014 | $1.16E−26$ |
| 23 | 2 | 0.0009 | 115.7405 | 1.08E−05 | 5.3576 | 0 |
| 24 | 1.6864 | 0.001 | 58.2724 | 9.08E−07 | 5.2665 | 0 |
| 25 | 0.4776 | 0.182 | 40.2556 | 7.38E−24 | 5.2213 | 0 |
| 26 | 1.9928 | 0.001 | 112.8824 | 1.03E−05 | 5.3553 | 0 |
| 27 | 1.8995 | 0.0389 | 107.3014 | 5.37E−06 | 5.3533 | 0 |
| 28 | 0.2306 | 0.001 | 40.401 | 2.23E−49 | 5.1854 | 0 |
| 29 | 2 | 0.001 | 115.7355 | 1.08E−05 | 5.3576 | 0 |
| 30 | 0.8333 | 0.0478 | 41.0522 | 1.09E−13 | 5.1989 | 0 |

monocrystalline PV cell (Shell SQ85) are presented in Table 6. Table 7 presents the results obtained by the run of the thin-film PV cell (Shell ST40) using the IEO.

Tables 5–7 reveal a small difference in the values of series resistance $R_S$ for both multicrystalline and monocrystalline. The I–V curve is strongly affected by the series resistance. According to the results, lower values of Rs lead the I–V characteristics to move farther from the axis with respect to the maximum point, while higher values of Rs lead the I–V characteristics to move closer to the axis as in Fig. 5. This tendency occurs naturally with the IEO parameter values.

**Table 7 Optimal variables for the single-diode model of thin-film photovoltaic cell, Shell ST40.**

| Run-No | Optimized parameters | | | Calculated parameters | | Error |
|---|---|---|---|---|---|---|
| | $a$ | $R_S$ (ohm) | $R_P$ (ohm) | $I_O$ (amp) | $I_{PV}$ (amp) | $E$ |
| 1 | 0.3722 | 0.0009 | 61.4825 | 1.90E−32 | 2.6800 | 0 |
| 2 | 2.0000 | 0.0055 | 80.7240 | 2.56E−06 | 2.6802 | 0 |
| 3 | 0.4832 | 0.1381 | 61.3438 | 4.47E−25 | 2.6860 | 0 |
| 4 | 1.8316 | 0.3748 | 86.4215 | 7.32E−07 | 2.6916 | $1.97E−31$ |
| 5 | 2.0000 | 0.7718 | 200.00 | 2.75E−06 | 2.6910 | 0 |
| 6 | 2.0000 | 0.0010 | 80.5100 | 2.56E−06 | 2.6800 | 0 |
| 7 | 0.6324 | 1.0000 | 60.8257 | 3.06E−19 | 2.7241 | 0 |
| 8 | 2.0000 | 0.7718 | 200.00 | 2.75E−06 | 2.6903 | 0 |
| 9 | 0.2588 | 0.2059 | 61.2756 | 1.69E−46 | 2.6890 | 0 |
| 10 | 0.1561 | 0.6689 | 60.8125 | 7.74E−77 | 2.7095 | 0 |
| 11 | 2.0000 | 0.0085 | 80.8260 | 2.56E−06 | 2.6803 | 0 |
| 12 | 1.0287 | 0.3286 | 62.2278 | 5.72E−12 | 2.6942 | 0 |
| 13 | 1.9993 | 0.1227 | 85.1764 | 2.56E−06 | 2.6839 | 0 |
| 14 | 0.1848 | 0.8890 | 60.5925 | 5.42E−65 | 2.7193 | $1.51E−24$ |
| 15 | 2.0000 | 0.6067 | 135.2353 | 2.69E−06 | 2.6920 | 0 |
| 16 | 1.0079 | 0.0010 | 61.8808 | 3.27E−12 | 2.6800 | 0 |
| 17 | 2.0000 | 0.5032 | 116.1444 | 2.66E−06 | 2.6916 | 0 |
| 18 | 2.0000 | 0.3980 | 103.5123 | 2.64E−06 | 2.6903 | $7.88E−31$ |
| 19 | 2.0000 | 0.0010 | 80.5112 | 2.56E−06 | 2.6800 | 0 |
| 20 | 2.0000 | 0.7127 | 168.6513 | 2.73E−06 | 2.6913 | $1.37E−24$ |
| 21 | 2.0000 | 0.4524 | 109.4282 | 2.65E−06 | 2.6911 | 0 |
| 22 | 2.0000 | 0.0009 | 80.5115 | 2.56E−06 | 2.6800 | 0 |
| 23 | 0.4069 | 0.0009 | 61.4825 | 1.03E−29 | 2.6800 | 0 |
| 24 | 0.1303 | 0.5279 | 60.9536 | 5.18E−92 | 2.7032 | 0 |
| 25 | 0.6593 | 0.5741 | 60.9933 | 1.79E−18 | 2.7052 | $8.27E−25$ |
| 26 | 0.4683 | 0.0021 | 61.4795 | 7.28E−26 | 2.6801 | $4.43E−29$ |
| 27 | 0.2947 | 0.3023 | 61.1791 | 6.99E−41 | 2.6932 | 0 |
| 28 | 0.9640 | 0.0308 | 61.7630 | 9.44E−13 | 2.6813 | $1.50E−26$ |
| 29 | 0.2048 | 0.5737 | 60.9078 | 1.14E−58 | 2.7052 | 0 |
| 30 | 0.4355 | 0.5503 | 60.9320 | 8.85E−28 | 2.7042 | 0 |

The difference between I–V characteristics can be attributed not only to different values of $R_S$ and $R_P$, but also to ideality factor of the diode, $a$. It is demonstrated that the values of all variables change significantly. However, every optimal solution can arrive at zero error for the three major points. All prior studies displayed one I–V characteristic that passed across the three main points. However, a PV cell characteristic was observed that is not normal when the designer concentrated on the three major points. For each cell, there is an operating point on the characteristic in which the performance is greatest. These points are referred as the major points. The algorithm arrived at the global optimal

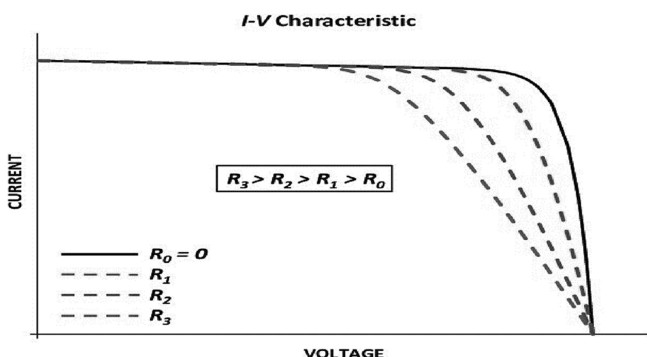

**Figure 5 Effects of the series resistance on the I–Vcharacteristic.**

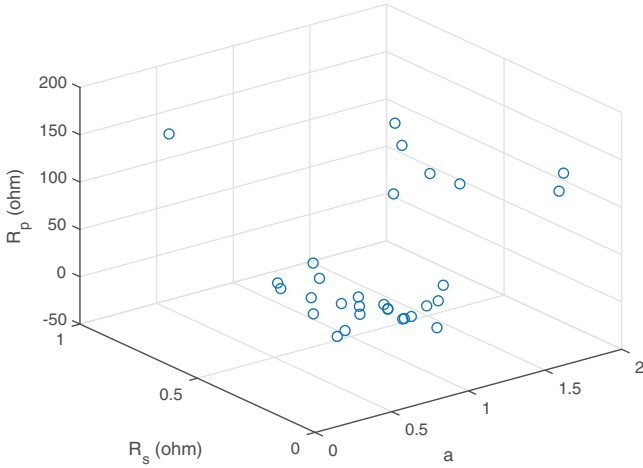

**Figure 6 Scatter plot of the optimalvalues (30 sets) for the single-diode model ofpolycrystalline cell KC200GT.**

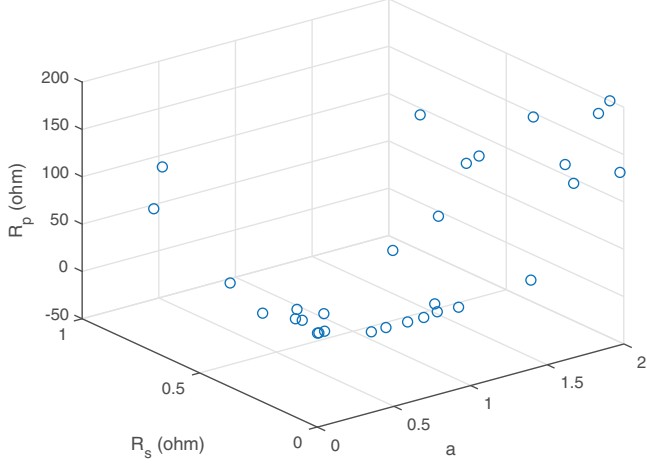

**Figure 7 Scatter plot of the optimalvalues for the single-diode model ofmonocrystalline cell SQ85.**

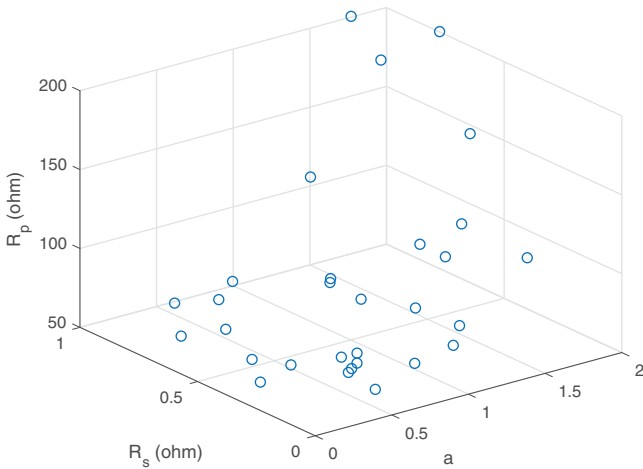

**Figure 8 Scatter plot of the optimalvalues for the single-diode model of thin-filmcell ST40.**

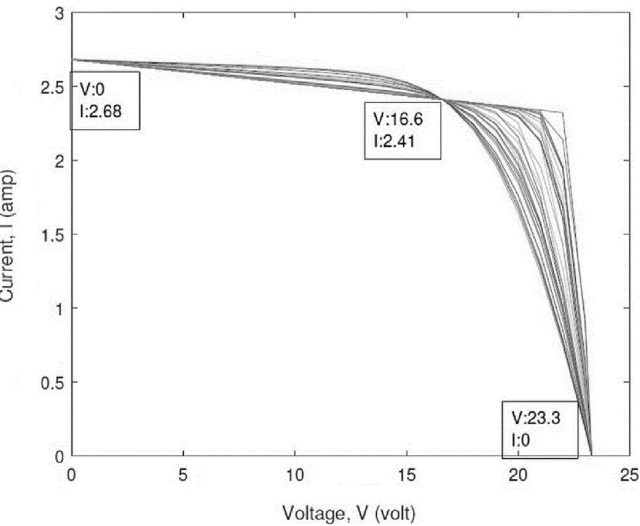

**Figure 9 Plot of the optimal solutions for the single-diodemodel of thin-film cell ST40.**

solution after 0.10 s with an error of zero value. This algorithm took less time to run compared to other metaheuristic algorithms.

A scatter plot of the optimal values (30 points) of these three parameters is provided in Figs. 6–8. It is showing that the value of each parameter changes significantly. This demonstrates that under the same parameter values $I_{PV}, I_O, a, R_S$ ~and~ $R_P$, the IEO algorithm has the highest efficiency for the experimental I–V data for every module.

As displayed in the previous figures, every curve passes through the three known points. A plot of the optimal solution is presented in Fig. 9. A difference in ($a$) is present approximately between 0.2 and 1.7, while that of ($R_p$) is approximately between 80 and 175 ohms.

**Table 8 Optimal variables for the double-diode model of polycrystalline photovoltaic cell, Kyocera KC200GT.**

| Run-No | Optimized parameters | | | | | Calculated parameters | | Error |
|---|---|---|---|---|---|---|---|---|
| | $a_1$ | $a_2$ | $R_S$ (ohm) | $R_P$ (ohm) | $I_{O1}$ (amp) | $I_{O2}$ (amp) | $I_{PV}$ (amp) | $E$ |
| 1 | 2.0000 | 1.5233 | 0.14386 | 67.3361 | 1.16E−07 | 3.21E−07 | 8.2275 | 0 |
| 2 | 1.8289 | 1.4930 | 0.2968 | 199.7722 | 1.00E−12 | 2.38E−07 | 8.2222 | 1.02E−29 |
| 3 | 2.0000 | 1.9995 | 0.0158 | 131.1378 | 1.00E−06 | 1.80E−05 | 8.2109 | 0 |
| 4 | 1.9915 | 1.2460 | 0.2805 | 61.5366 | 1.00E−12 | 7.34E−09 | 8.2474 | 0 |
| 5 | 1.9999 | 2.0000 | 0.0010 | 117.3639 | 1.00E−06 | 1.79E−05 | 8.2101 | 0 |
| 6 | 1.8059 | 1.5498 | 0.0010 | 50.0000 | 2.15E−07 | 4.013E−07 | 8.2102 | 0 |
| 7 | 1.9687 | 1.6440 | 0.0010 | 55.3115 | 1.00E−12 | 1.11E−06 | 8.2101 | 0 |
| 8 | 1.6413 | 2.0000 | 0.0012 | 117.0978 | 6.92E−09 | 1.89E−05 | 8.2101 | 0 |
| 9 | 2.0000 | 1.4957 | 0.0412 | 50.0000 | 1.00E−12 | 2.31E−07 | 8.2168 | 0 |
| 10 | 1.9717 | 1.9995 | 0.0010 | 117.1425 | 3.52E−09 | 1.89E−05 | 8.2101 | 0 |
| 11 | 2.0000 | 0.9102 | 0.4325 | 53.7084 | 4.62E−07 | 3.34E−12 | 8.2761 | 1.26E−29 |
| 12 | 1.6973 | 1.1087 | 0.2937 | 50.0000 | 2.34E−09 | 5.51E−10 | 8.2582 | 0 |
| 13 | 1.9769 | 2.0000 | 0.0027 | 118.7968 | 1.51E−12 | 1.90E−05 | 8.2102 | 0 |
| 14 | 2.0000 | 1.0519 | 0.3943 | 61.4608 | 1.00E−12 | 1.59E−10 | 8.2627 | 7.61E−28 |
| 15 | 1.8233 | 1.5153 | 0.0049 | 50.0000 | 1.00E−06 | 2.32E−07 | 8.2108 | 0 |
| 16 | 1.9087 | 1.5505 | 0.0010 | 50.0000 | 2.73E−07 | 4.13E−07 | 8.2102 | 7.89E−31 |
| 17 | 1.5751 | 2.0000 | 0.0141 | 129.5679 | 1.00E−12 | 1.91E−05 | 8.2109 | 0 |
| 18 | 1.9727 | 1.2646 | 0.1699 | 50.0000 | 1.00E−06 | 9.14E−09 | 8.2379 | 0 |
| 19 | 2.0000 | 2.0000 | 0.0012 | 117.5239 | 9.77E−07 | 1.80E−05 | 8.2101 | 0 |
| 20 | 1.9745 | 1.9999 | 0.0327 | 153.4717 | 1.00E−12 | 1.91E−05 | 8.2117 | 0 |
| 21 | 1.9170 | 2.0000 | 0.0010 | 117.3639 | 1.00E−12 | 1.89E−05 | 8.2101 | 7.13E−28 |
| 22 | 1.9999 | 1.5504 | 0.0012 | 50.0009 | 3.68E−07 | 4.15E−07 | 8.2102 | 0 |
| 23 | 1.9920 | 2.0000 | 0.0010 | 117.3640 | 1.00E−12 | 1.89E−05 | 8.2101 | 0 |
| 24 | 1.6250 | 1.4812 | 0.2960 | 200.0000 | 1.90E−07 | 1.67E−07 | 8.2222 | 0 |
| 25 | 1.5044 | 1.3441 | 0.1288 | 50.5678 | 9.34E−08 | 2.08E−08 | 8.2309 | 0 |
| 26 | 1.3900 | 1.9999 | 0.0010 | 117.3636 | 1.00E−12 | 1.89E−05 | 8.2101 | 0 |
| 27 | 1.9999 | 2.0000 | 0.0010 | 117.3639 | 7.48E−09 | 1.89E−05 | 8.2101 | 0 |
| 28 | 2.0000 | 1.9947 | 0.0320 | 148.5751 | 2.98E−09 | 1.85E−05 | 8.2118 | 0 |
| 29 | 1.9904 | 1.5530 | 0.0034 | 50.0000 | 1.39E−08 | 4.36E−07 | 8.2106 | 3.15E−30 |
| 30 | 1.9982 | 0.5695 | 0.6635 | 55.6877 | 1.53E−09 | 1.41E−19 | 8.3078 | 7.88E−31 |

## Results of the double diode model

Using the (same parameter values) $I_{ph}$, $I_{o1}$, $I_{o2}$, $a_1$, $a_2$, $R_S$, and $R_P$, IEO is more consistent with the experimental I–V data of all types of modules. The estimation of the parameters of SCs based on the DDM has high precision, and the optimization of parameters is implemented for the DDM of different types. A total of five parameters are optimized for the DDM ($a_1$, $a_2$, $R_S$, $R_P$, $I_{O1}$), while two sustainable parameters are calculated (IO2, IPV) for their relation with the optimized parameters. The optimized parameters in various runs of the IEO algorithm for the DDM polycrystalline, KC200GT cell are listed in

**Table 9 Optimal variables for the double-diode model of monocrystalline Shell SQ85.**

| Run-No | Optimized parameters | | | | | Calculated parameters | | | Error |
|---|---|---|---|---|---|---|---|---|---|
| | $a_1$ | $a_2$ | $R_S$ (ohm) | $R_P$ (ohm) | $I_{O1}$ (amp) | $I_{O2}$ (amp) | $I_{PV}$ (amp) | $E$ | |
| 1 | 2.0000 | 1.3534 | 0.1210 | 50.0000 | 7.84E−07 | 1.82E−08 | 5.4632 | 0 | |
| 2 | 2.0000 | 2.0000 | 0.0010 | 117.3655 | 1.00E−12 | 1.08E−05 | 5.4500 | 0 | |
| 3 | 2.0000 | 1.5570 | 0.0010 | 50.0000 | 1.00E−12 | 2.47E−07 | 5.4501 | 0 | |
| 4 | 2.0000 | 1.7412 | 0.0261 | 69.0104 | 4.49E−07 | 1.43E−06 | 5.4521 | 0 | |
| 5 | 1.9727 | 2.0000 | 0.0010 | 117.0170 | 8.39E−07 | 9.76E−06 | 5.4500 | 0 | |
| 6 | 1.9675 | 2.0000 | 0.0010 | 117.3639 | 1.00E−12 | 1.08E−05 | 5.4500 | 7.88E−31 | |
| 7 | 2.0000 | 1.5394 | 0.0010 | 50.0000 | 1.00E−06 | 1.84E−07 | 5.4501 | 0 | |
| 8 | 1.8821 | 2.0000 | 0.0010 | 114.3371 | 9.90E−07 | 8.51E−06 | 5.4500 | 0 | |
| 9 | 1.7597 | 1.9999 | 0.0056 | 119.4022 | 1.22E−07 | 1.00E−05 | 5.4503 | 0 | |
| 10 | 1.7776 | 0.6201 | 0.5195 | 50.0000 | 5.62E−07 | 1.64E−18 | 5.5066 | 3.16E−28 | |
| 11 | 1.9716 | 1.8592 | 0.0010 | 80.0694 | 1.00E−06 | 3.48E−06 | 5.4501 | 0 | |
| 12 | 1.5348 | 1.5560 | 0.0016 | 50.0000 | 1.05E−12 | 2.44E−07 | 5.4502 | 0 | |
| 13 | 1.9755 | 1.2397 | 0.2009 | 50.0005 | 3.33E−07 | 3.21E−09 | 5.4719 | 0 | |
| 14 | 1.9170 | 2.0000 | 0.0010 | 117.3642 | 1.00E−12 | 1.08E−05 | 5.4500 | 0 | |
| 15 | 1.8099 | 1.2020 | 0.4245 | 158.2137 | 1.00E−12 | 1.82E−09 | 5.4646 | 0 | |
| 16 | 1.5581 | 1.5565 | 0.0010 | 50.0000 | 3.09E−07 | 5.85E−08 | 5.4501 | 0 | |
| 17 | 1.9975 | 0.5000 | 0.6887 | 50.0000 | 1.00E−12 | 8.92E−23 | 5.5251 | 0 | |
| 18 | 1.7492 | 2.0000 | 0.0010 | 117.3639 | 1.00E−12 | 1.08E−05 | 5.4500 | 6.19E−25 | |
| 19 | 1.9707 | 2.000 | 0 0.0013 | 117.5777 | 1.00E−12 | 1.08E−05 | 5.4501 | 0 | |
| 20 | 1.9769 | 2.0000 | 0.0237 | 140.7349 | 1.33E−07 | 1.07E−05 | 5.4509 | 0 | |
| 21 | 1.8546 | 1.6003 | 0.2212 | 145.2258 | 2.37E−07 | 3.87E−07 | 5.4583 | 7.88E−31 | |
| 22 | 1.8666 | 1.2806 | 0.2578 | 66.6718 | 8.05E−07 | 5.39E−09 | 5.4711 | 0 | |
| 23 | 1.7608 | 1.9859 | 0.0023 | 112.7487 | 1.00E−12 | 9.80E−06 | 5.4501 | 0 | |
| 24 | 2.0000 | 2.0000 | 0.0010 | 117.3639 | 3.79E−07 | 1.04E−05 | 5.4500 | 0 | |
| 25 | 2.0000 | 2.0000 | 0.0546 | 200.0000 | 1.00E−06 | 9.93E−06 | 5.4515 | 1.97E−29 | |
| 26 | 1.9668 | 1.9200 | 0.0095 | 96.2587 | 5.21E−07 | 5.82E−06 | 5.4505 | 0 | |
| 27 | 1.8934 | 2.0000 | 0.0037 | 119.6661 | 3.89E−09 | 1.08E−05 | 5.4502 | 0 | |
| 28 | 1.9697 | 1.9996 | 0.0010 | 116.9501 | 5.45E−07 | 1.01E−05 | 5.4500 | 4.65E−25 | |
| 29 | 2.0000 | 2.0000 | 0.0010 | 117.3639 | 1.03E−12 | 1.08E−05 | 5.4500 | 0 | |
| 30 | 1.9478 | 1.5355 | 0.0010 | 50.0000 | 1.00E−06 | 1.68E−07 | 5.4501 | 0 | |

Table 8, SQ85 monocrystalline dish are listed in Table 9 and thin cells, shell ST40 are displayed in Table 10. For the SDM, the optimized parameters resulted in zero error at the three main points. The 30 I–V characteristics resulting from various combinations of the variables are presented in Fig. 9. A plot of the optimal solution is presented in Fig. 10. All the curves passed through the open circuit, short circuit, and maximum power points. To easily analyze the optimized variables, standardized parameter values are presented in a parallel coordinate graph in Fig. 11. The value of $R_P$ differs widely. However, $I_{O1}$ ranges from approximately 0.01 to 0.9p.u. The final value of the diode saturation current is very low, with an average of only $\mu A$

Table 10 Optimal variables for the double-diode model of thin-film photovoltaic cell, Shell ST40.

| Run-No | Optimized parameters | | | | | Calculated parameters | | | Error |
| --- | --- | --- | --- | --- | --- | --- | --- | --- | --- |
| | $a_1$ | $a_2$ | $R_S$ (ohm) | $R_P$ (ohm) | $I_{O1}$ (amp) | $I_{O2}$ (amp) | $I_{PV}$ (amp) | $E$ | |
| 1 | 1.9497 | 0.8051 | 0.0010 | 61.5456 | 1.00E−12 | 3.39E−15 | 2.6800 | 0 | |
| 2 | 2.0000 | 1.9998 | 0.7719 | 199.9620 | 1.00E−06 | 1.75E−06 | 2.6904 | 7.96E−29 | |
| 3 | 2.0000 | 0.6828 | 0.1757 | 61.7124 | 1.68E−07 | 6.97E−18 | 2.6876 | 0 | |
| 4 | 2.0000 | 1.7425 | 1.0000 | 200.0000 | 1.00E−06 | 2.30E−07 | 2.6934 | 0 | |
| 5 | 1.9999 | 0.8567 | 0.0518 | 61.5676 | 3.69E−09 | 2.65E−14 | 2.6823 | 0 | |
| 6 | 1.9495 | 2.0000 | 0.2125 | 89.6850 | 1.00E−12 | 2.59E−06 | 2.6864 | 0 | |
| 7 | 1.7032 | 0.5000 | 0.4038 | 64.0517 | 1.79E−07 | 6.46E−25 | 2.6969 | 0 | |
| 8 | 1.6443 | 1.5964 | 0.4139 | 75.9185 | 2.32E−07 | 6.12E−08 | 2.6946 | 1.84E−25 | |
| 9 | 1.9999 | 0.5185 | 0.0010 | 63.3347 | 1.00E−06 | 1.28E−23 | 2.6800 | 0 | |
| 10 | 1.9396 | 2.0000 | 0.7815 | 199.6071 | 1.00E−06 | 1.22E−06 | 2.6905 | 1.77E−30 | |
| 11 | 2.0000 | 0.5958 | 0.5694 | 63.31576 | 6.10E−07 | 1.60E−20 | 2.7041 | 4.20E−27 | |
| 12 | 1.8235 | 0.5501 | 1.0000 | 60.5959 | 1.00E−12 | 4.59E−22 | 2.7242 | 0 | |
| 13 | 1.9419 | 0.5000 | 0.8368 | 60.6656 | 1.00E−12 | 3.05E−24 | 2.7169 | 0 | |
| 14 | 1.9999 | 0.5000 | 0.0010 | 63.3348 | 1.00E−06 | 1.80E−24 | 2.6800 | 0 | |
| 15 | 1.7237 | 0.5000 | 0.7316 | 60.7619 | 1.00E−12 | 3.05E−24 | 2.7123 | 0 | |
| 16 | 1.7183 | 0.6149 | 1.0000 | 62.3562 | 4.42E−08 | 7.48E−20 | 2.7229 | 0 | |
| 17 | 1.7775 | 2.0000 | 0.8158 | 200.0000 | 3.65E−07 | 7.10E−07 | 2.6909 | 3.86E−29 | |
| 18 | 1.8125 | 1.6049 | 0.9979 | 132.9710 | 3.09E−07 | 4.81E−08 | 2.7001 | 0 | |
| 19 | 1.4287 | 0.5322 | 0.7106 | 71.8892 | 4.00E−08 | 2.46E−22 | 2.7065 | 1.13E−28 | |
| 20 | 1.9741 | 0.5007 | 0.0049 | 62.6031 | 5.44E−07 | 2.40E−24 | 2.6802 | 0 | |
| 21 | 1.9138 | 1.9787 | 0.7749 | 188.6456 | 1.21E−09 | 2.37E−06 | 2.6910 | 1.77E−30 | |
| 22 | 2.0000 | 0.5000 | 0.2572 | 61.2253 | 1.00E−12 | 3.02E−24 | 2.6913 | 0 | |
| 23 | 1.9783 | 0.7001 | 0.0021 | 61.4963 | 1.01E−12 | 2.02E−17 | 2.6801 | 0 | |
| 24 | 2.0000 | 0.5000 | 1.0000 | 60.5306 | 1.00E−12 | 3.06E−24 | 2.7242 | 0 | |
| 25 | 1.9991 | 0.5000 | 0.4302 | 61.0538 | 1.00E−12 | 3.03E−24 | 2.6989 | 0 | |
| 26 | 1.4757 | 0.7693 | 0.0010 | 61.5234 | 1.06E−12 | 6.90E−16 | 2.6800 | 0 | |
| 27 | 1.9429 | 1.3129 | 0.6557 | 71.6179 | 3.02E−07 | 1.57E−09 | 2.7045 | 0 | |
| 28 | 1.9632 | 2.0000 | 0.0010 | 80.3427 | 8.82E−07 | 1.42E−06 | 2.6800 | 0 | |
| 29 | 2.0000 | 0.5000 | 0.1255 | 62.7184 | 6.31E−07 | 2.26E−24 | 2.6854 | 0 | |
| 30 | 1.7438 | 0.5000 | 0.1682 | 63.5249 | 2.48E−07 | 7.51E−25 | 2.6871 | 0 | |

To summarize, regardless of the type of the PV cell or the PV parameter extraction model used, we cannot obtain one I–V characteristic with details of short circuits, maximum power, and open circuit points. A total of three point optimizations can reveal various characteristics in several tests. 'Simulation Results and Analysis' confirms the effectiveness of the IEO algorithm in estimating the parameters of the SC, taking into account the SDM and DDM. In addition, the SDM used three types of PV modules to estimate the parameters of the PV model.

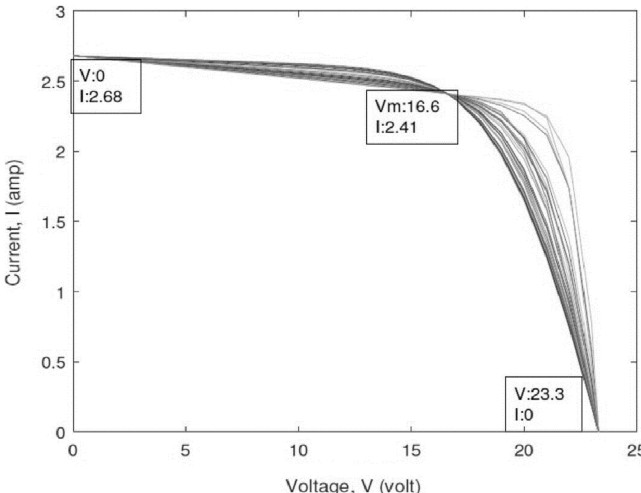

**Figure 10 Plot of the optimal solutions for the double-diodemodel of thin film ST40.**

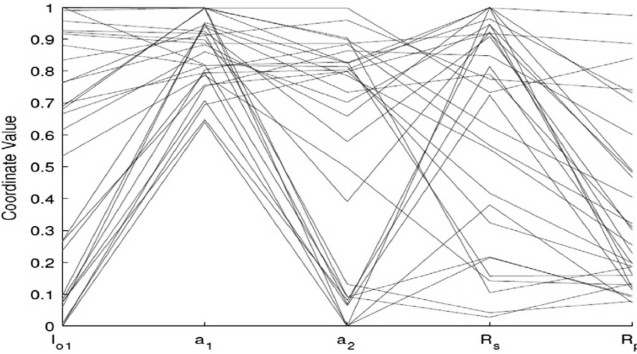

**Figure 11 Parallel coordinate plot of the optimal solutions forthe double-diode model of thin-film ST40.**

# RESULTS OF COMPARING THE EFFECIENCY OF THE IEO WITH OTHER METAHEURISTIC ALGORITHMS

The results of SC parameter estimation are compared with the results of other metaheuristic algorithms to evaluate the efficiency of applying the IEO algorithm.

## The single diode model

This subsection presents a comparison between the solutions generated by the IEO algorithm and other algorithms that have been applied under the same conditions. The optimal error values obtained by the IEO was zero, as presented in Table 11. The IEO achieves the best results among all the six algorithms tested, followed by SCA, while the GA achieves the poorest results.

## The double diode model

This subsection provides a comparison between the solutions generated by the IEO algorithm and other algorithms for the DDM. Table 12 indicates that the best error values

**Table 11 Comparison of various algorithms and IEO for the single-diode model.**

| Method | $I_{ph}(A)$ | $I_O$ | $a$ | $R_S$ | $R_P$ | Error | Average | Std |
|---|---|---|---|---|---|---|---|---|
| GA | 0.7619 | 0.8087 | 1.5751 | 0.0299 | 52.3729 | 0.01908 | 0.07834 | 9.20E−04 |
| PSO | 8.2305 | 3.09E−10 | 1.0802 | 0.2422 | 96.8938 | 0.0069 | 0.0059 | 3.48E−05 |
| WOA | 8.2255 | 8.05E−11 | 1.022 | 0.3253 | 172.4225 | 8.50E−15 | 4.17E−16 | 1.78E−15 |
| GWO | 8.2109 | 5.78E−22 | 0.5084 | 0.005 | 43.8802 | 9.23E−10 | 2.29E−11 | 5.49E−11 |
| SCA | 8.2685 | 5.04E−27 | 0.4137 | 0.313 | 43.894 | 6.66E−14 | 1.73E−09 | 2.44E−09 |
| MVO | 8.2106 | 1.25E−08 | 1.279 | 0.00524 | 64.3523 | 1.02E−07 | 3.91E−10 | 1.70E−09 |
| EO | 8.2133 | 2.23E−08 | 1.3159 | 0.0292 | 72.1747 | 1.81E−23 | 5.52E−17 | 3.02E−16 |
| **IEO** | **8.2178** | **1.02E−08** | **1.266** | **0.0709** | **74.3245** | **0** | **3.46E−29** | **1.89E−28** |

**Note:**
Best values shown in bold.

**Table 12 Comparison of the results of various algorithms and the IEO for the double-diode model.**

| Method | $I_{ph}(A)$ | $I_{O1}$ | $I_{O2}$ | $a_1$ | $a_2$ | $R_S$ | $R_P$ | Error | Avg | Std |
|---|---|---|---|---|---|---|---|---|---|---|
| GA | 0.8207 | 3.47E−09 | 1.13E−09 | 1.38 | 1.15 | 0.2633 | 92.64 | 0.0670 | 0.065 | 0.009 |
| PSO | 4.2023 | 2.04E−09 | 1.17E−10 | 1.21 | 1.07 | 0.17804 | 104.36 | 0.0075 | 0.006 | 8.9E−04 |
| WOA | 8.2125 | 3.41E−07 | 2.79E−11 | 1.82 | 0.99 | 0.0152 | 50.24 | 2.23E−14 | 1.2E−15 | 3.9E−15 |
| GWO | 8.2325 | 6.15E−07 | 1.08E−12 | 2 | 0.88 | 0.1416 | 51.58 | 1.08E−11 | 7.9E−10 | 4.2E−09 |
| SCA | 8.2101 | 1.42E−12 | 7.19E−06 | 1.22 | 1.67 | 0.0042 | 200 | 2.78E−08 | 4.5E−09 | 1.1E−08 |
| MVO | 8.2122 | 4.04E−07 | 3.97E−07 | 2 | 1.54 | 0.0403 | 149.29 | 1.77E−10 | 1.2E−09 | 3.1E−09 |
| EO | 8.2153 | 5.53E−07 | 3.16E−08 | 1.95 | 1.43 | 0.0964 | 149.79 | 5.28E−21 | 4.5E−18 | 2.4E−17 |
| **IEO** | **8.2108** | **1.00E−06** | **2.32E−07** | **1.82** | **1.52** | **0.0049** | **50** | **0** | **2.8E−22** | **1.5E−21** |

**Note:**
Best values shown in bold.

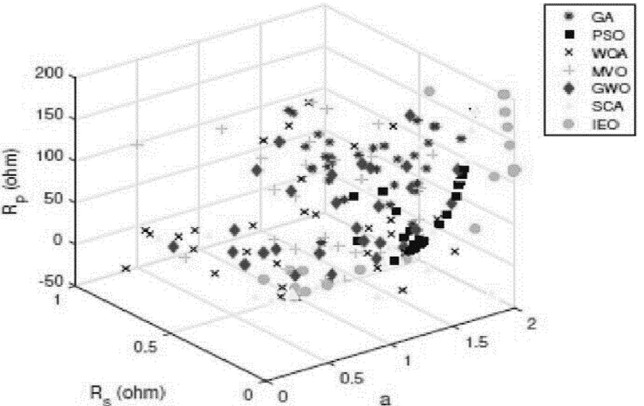

**Figure 12 Scatter plot of the optimal results for the single-diodemodel of KC200GT cell using different algorithms.**

obtained by the IEO were zero. The IEO thus achieved the best results of the six algorithms tested, followed by SCA. Whereas, GA achieved the poorest results.

To sum up the obtained results, Tables 11 and 12 indicate that PSO produces better results than GA. In addition, GWO generates fewer errors than PSO, GA, WOA, SCA, and

MVO. However, the IEO results are superior to those of all other approaches. Fig. 12 presents a comparison between the optimal results for the SDM of the KC200GT cell using different methods.

## CONCLUSIONS AND FUTURE WORK

Developing an accurate model of the PV system is an important and challenging task. To obtain an accurate model, it is necessary to determine the optimal parameters for the PV model using an effective optimization technique. The IEO algorithm is a recent optimization technique that is used for parameter extraction of the PV modules for the SDM and DDM. The IEO algorithm has various advantages, such as solution accuracy, convergence speed, and balance between analysis and exploitation. In this paper, the PV generator is modeled using the SDM and DDM, and the parameters of each model were optimally determined using the IEO algorithm. The IEO has been compared to the following six metaheuristic algorithms: GA, PSO, SCA, WOA, MVO, and GWO. The results achieved using the IEO are accurate than those achieved by the other methods. The IEO could be a good candidate for solving the optimization problem of the SC systems. In future work, the IEO can be applied to identify the PV parameters for the multi-dimension diode and multi-diode models. Moreover, it can also be used for calculating the current-voltage characteristic of the multi-diode models.

## ABBREVIATIONS

| | |
|---|---|
| **PV** | Photovoltaic |
| **IEO** | Improved Equilibrium Optimizer algorithm |
| **EO** | Equilibrium Optimizer algorithm |
| **BL** | Opposition Based Learning |
| **SDM** | single diode model |
| **DDM** | double diode model |
| **RESs** | renewable energy sources |
| **SCs** | system uses solar cells |
| $V_{oc}$ | open circuit voltage |
| **Vmpp** | voltage at the point of maximum power |
| $I_{mpp}$ | current at the point of maximum power |
| $I_{sc}$ | short-circuit current |
| $P_{mpp}$ | maximum power |
| **Kv** | voltage temperature coefficients |
| **Ki** | current temperature coefficients |
| $I_o$ | the reverse saturation current |
| **Ipv** | photogenerated current |
| **a** | non-physical ideality factor of the diode |
| $R_p$ | shunt resistance |
| $R_s$ | series resistant |
| **GA** | a genetic algorithm |

| PSO | particle swarm optimization |
|---|---|
| SA | simulated annealing |
| GWO | gray wolf optimizer |
| FPA | flower pollination algorithm |
| MSSO | simplified swarm optimization algorithm |
| $I_D$ | The diode current |
| $I_{o,cell}$ | The reverse saturation current of the diode |
| $N_c$ | represents the number of cells |
| K | Boltzmann constant |
| q | charge of an electron |
| T | The temperature (in Kelvin) |
| $C^{initial}$ | The initial condensation of the ith particle |
| $C_{max}$ | The maximum of the dimensions |
| $C_{min}$ | The minimum of the dimensions |
| n | the particle number representing a population |
| G | key expression of the proposal technique |
| WOA | whale optimizer algorithm |
| SCA | since-cosine algorithm |
| MVO | Multi-Verse Optimizer |
| ABC | artificial bee colony |
| WDO | Wind Driven Optimization |
| SCE | Sister chromatid exchange |
| ImCSA | improved cuckoo search algorithm |
| ISCA | Interview Schedule For Children and Adolescents |
| SMA | Superior Mesenteric artery |
| COA | Coyote optimization algorithm |
| TLO | Teaching-learning-based optimization |

### Funding
The authors received no funding for this work.

### Competing Interests
The authors declare that they have no competing interests.

### Author Contributions
- Essam H. Houssein conceived and designed the experiments, performed the experiments, analyzed the data, prepared figures and/or tables, authored or reviewed drafts of the paper, and approved the final draft.

- Gamela Nageh conceived and designed the experiments, performed the experiments, analyzed the data, performed the computation work, prepared figures and/or tables, authored or reviewed drafts of the paper, and approved the final draft.
- Mohamed Abd Elaziz conceived and designed the experiments, performed the experiments, analyzed the data, prepared figures and/or tables, authored or reviewed drafts of the paper, and approved the final draft.
- Eman Younis conceived and designed the experiments, performed the experiments, analyzed the data, prepared figures and/or tables, authored or reviewed drafts of the paper, and approved the final draft.

## Data Availability

The improved code for equilibrium optimization using opposition-based learning and the code to generate simulated datasets used to test the code is available in the Supplemental Files.

## Supplemental Information

Supplemental information for this article can be found online at http://dx.doi.org/10.7717/peerj-cs.708#supplemental-information.

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
