# Peer review of "An efficient Equilibrium Optimizer for parameters identification of photovoltaic modules"

_PeerJ Computer Science, doi:10.7717/peerj-cs.708_

## Round 0.1 · original submission · Major Revisions

Based on the criticisms of this works, authors should carefully address all the comments and provide detailed and clear indications of how authors address the comments and implement these comments into the revised manuscript.

Reviewer 1 ·

Basic reporting

Unfortunately, I found this manuscript very difficult to follow and dotted with many unclear sentences and grammatical errors.

Statements made in the manuscript are not adequately supported by relevant references.

Specific examples are provided under general comments to authors

Experimental design

The research question is well defined. However, I am not sure the actual work contributes to solving it.

Although the methods are described in detail, I found the description of the algorithm very difficult to follow.

Specific examples are provided under general comments to authors

Validity of the findings

The findings seem to be validated. However, as noted above, I don't see how they contribute to the research question.

Specific examples are provided under general comments to authors

Additional comments

This manuscript describes the application of the IEO algorithm enhanced by opposition-based learning, for the optimization of certain parameters related to the performances of solar cells. Three different types of cells are considered, each in either a single diode model or in a double diode model configuration. It is concluded that the algorithm is consistently able to identify the optimal set of parameters that leads to the smallest error (in comparison with other algorithms) between experimental and calculated I-V values across three points.

While solar cells are certainly important as a source of green renewable energy and while their performances clearly need to be optimized, I fail to see how this algorithm contributes to this endeavor.

In addition, I, unfortunately, found this manuscript very difficult to follow and dotted with many unclear sentences and grammatical errors. More specific concerns are listed below.

Major points
1. In the abstract as well as in line 50, the authors suggest that determining the values of unknown parameters is important for optimizing solar cells. However, this paper deals with finding optimal values for a set of intrinsic solar cell parameters that cannot be directly controlled and using them to accurately calculate other intrinsic parameters. Can the authors explain how this has anything to do with the optimization of solar cell performances?
2. On line 57 the authors suggest that certain parameters characterizing SDM are not provided by solar cell manufacturers and determining their values is challenging. Is this because solar cells manufacturers neglect to provide these values or because they don’t know them? Looking at the results later on in the manuscript, it seems that many combinations of these parameters are all in accord with solar cell global parameters. Given that, perhaps these unknown parameters do not have unique values in which case, how can they be determined?
3. The two paragraphs starting on lines 74 and 85 seems to discuss the same topic, namely, metaheuristic optimization algorithms. Why not combine them?
4. Line 95: The statement that the methods described in the previous paragraphs use large computational resources and lead to high error rates must be supported by several references.
5. Line 100: I am not sure what the authors mean by an accurate PV model.
6. EO algorithm section: Unfortunately, I find the explanation of the EO algorithm lacking and very unclear to the point that it is difficult to understand it. For example, what is the initial population mentioned in line 158? What is the concentration mentioned in line 159? What is the condensation mentioned in line 161? What are the dimensions mentioned in line 162? What is the last case of convergence mentioned in line 165? What are the wolves mentioned in line 170? Perhaps an overview of the algorithm will help to clarify these points.
7. Line 204: The multiple statements in this section regarding the improved performances of the OBL algorithm should be supported by references.
8. Line 242: What are the differences between the different runs of the IEO algorithm?
9. Table 3: How were the ranges of each of the optimized parameters determined? This is especially important since in some cases, during the optimization, some of the parameters reached their boundary values (e.g., the Rs in entry 23 in Table 4 or the a parameter in many entries of Table 5).
10. Line 263: It seems to me that Table 5 presents all the results for the monocrystalline cell and not just the best ones.
11. Line 266: A small difference relative to what? It seems that the differences between the Rs values across different runs of the same system are much larger that the differences in this parameter between different systems.
12. Line 268: I don’t understand how the results say anything about the relationship between Rs values and the I-V curve.
13. Line 275: This sentence is completely unclear to me.
14. Line 279: Why is the scattering observed in Figures 5-7 indicative that the IEO algorithm has the highest efficiency for the experimental I-V data? Also, highest efficiency in comparison to what?
15. Figures 8 and 9: My understanding is that the algorithm was used to optimize parameter values so as to reduce the errors at three specific points on the I-V curves. How where then complete I-V curves constructed?
16. Line 306: The optimization problems presented in this manuscript are pretty simple and encompass only three variables in the case of SDM and only five in the case of DDM. Thus, I find it strange that none of the other methods were able to perform well in these cases. This is especially true for the GA algorithm that provided very poor results. The authors should provide details on how the other methods were used in order to try and figure out the reason for their poor performances.
17. The manuscript is dotted with many grammatical errors. Some, but not all, are given below. I would recommend that the manuscript is reviewed by a native English speaker.

Minor points
1. Page 1, line 15: Change “PV system that include to define” to “PV system to define”.
2. Line 23: Change “inspired from” to “inspired by”.
3. Line 25: Change “in the estimating” to “in estimating”.
4. Line 27: Change “is excellent” to “is superior”.
5. Line 49: The sentence containing “…five unknown parameters unknown that unknown while the…” should be re-written.
6. Line 90: Change “the” to “The” at the beginning of the sentence.
7. Line 108: Change “which inspire” to “which is inspired”.
8. Line 110: Change “variable” to “variables”.
9. Line 111: Change “generate less error” to “generate smaller errors”.
10. The paragraph starting at line 113 contains several references to numbered sections. However, none of the sections in this manuscript are numbered.
11. Line 117: Change “conclusion” to “conclusions”.
12. Line 119: What is the meaning of “can be approaching”?
13. Line 200: Why is this section marked as “1”?
14. Line 201: Change “we” to “We”.
15. Line 201: Change “closed” to “closest”.
16. Line 203: Change “given” to “gives”.
17. Line 205: What is the meaning of the question mark?
18. Line 217: Change “mult-idimensions” to “multi-dimensions”.
19. Line 228: Which section?
20. Line 231: Which section?

Reviewer 2 ·

Basic reporting

good

Experimental design

not required

Validity of the findings

well done

Additional comments

This paper presents A Parameter estimation of photovoltaic modules using an improved equilibrium optimizer algorithm using opposition based learning. Generally, the paper is well written and the presentation is acceptable; the reviewer has the following comments and questions for the authors to address:

1. Abstract need to be improved to highlight the contribution and the significant findings
2. More critical review should be added to the paper with reference to the latest published papers such as “Modeling of PV system and parameter extraction based on experimental data: Review and investigation” and “Parameter Extraction of Solar Cell Using Penalty Based Differential Evolution”
3. The title of the paper needs to be changed to be more specific
4. Discussion on the limitation of the proposed method should be added
5. The presentation of the results should be improved (color )
6. Detail comparison with the most recent topologies is required in the form of a table

Reviewer 3 ·

Basic reporting

1. In general, this paper is well written and contribute to the new knowledge in this field.

Experimental design

1. What are the constraints considered for solving the objective function? It should clearly define and explain the methodology and summarize in the flowchart i.e in Figure 4.

2. Section: Comparative Study with Other Metaheuristics -
How do you compare with other algorithms?
Have you run the algorithms or compare them with the reported results by these algorithms?
How do you set the parameter setting for these algorithms for fair comparison? How to ensure the comparison study have been done properly / fair in term of parameter setting and so on.. most of the Heuristics algorithms are affected by the parameter setting i.e population size, max iteration, and others parameters. it should clearly explain in this paper.

Validity of the findings

1. Error in Table 7, 8, 9 are zero (0). It recommended to display the value up to set decimal places such 4 decimal places.

2. Based on table 7 (Comparison), the reported errors are in decimal places. please revise for a fair comparison. What do you mean by Zero for IEO. The error of CSA 666E-14 also can describe as Zero (0)

Additional comments

This paper is suitable for publication after the authors revise the paper according to the comments/suggestions for improving the quality of the paper.

1. In general, this paper is well written and contribute to the new knowledge in this field.

Methodology
==========
1. What are the constraints considered for solving the objective function? It should clearly define and explain the methodology and summarize in the flowchart i.e in Figure 4.

2. Section: Comparative Study with Other Metaheuristics -
How do you compare with other algorithms?
Have you run the algorithms or compare them with the reported results by these algorithms?
How do you set the parameter setting for these algorithms for fair comparison? How to ensure the comparison study have been done properly / fair in term of parameter setting and so on.. most of the Heuristics algorithms are affected by the parameter setting i.e population size, max iteration, and others parameters. it should clearly explain in this paper.


Results and Discussion
==================
1. Error in Table 7, 8, 9 are zero (0). It recommended to display the value up to set decimal places such 4 decimal places.

2. Based on table 7 (Comparison), the reported errors are in decimal places. please revise for a fair comparison. What do you mean by Zero for IEO. The error of CSA 666E-14 also can describe as Zero (0)

Thanks

---

## Round 0.2 · Minor Revisions

Date: 20/07/2021

After the first round of revisions, reviewers felt that the work still requires a minor revision. Therefore, authors should carefully address and implement the comments from the reviewers.

The following are some other comments, authors should consider.

1. The quality of the figures in the manuscript should improve significantly,

2. Some grammatical and typo mistakes in the revised manuscript should be carefully corrected. Perhaps authors should ask a fluent English speaker to proofread this manuscript before the next submission.

Please address these and the concerns and criticisms of the reviewers detailed at the bottom of this letter and resubmit your article once you have updated it accordingly.

Reviewer 1 ·

Basic reporting

Has considerably improved and is now very good. Still, there are a few errors which are listed below in the general comment section.

Experimental design

Very good

Validity of the findings

The findings are valid.

Additional comments

The authors have considerably improved the level of the manuscript and have satisfactorily addressed almost all of my comments to the original submission. However, there are still few remaining points that should be addressed prior to publication:

Major points
1. Many combinations of the optimized parameters led to the same error (i.e., 0). Does this fact has any practical meaning in the context of designing new SCs? Is there any way to further rank these combinations?
2. Page 11, line 230: Despite the authors’ answer, I still don’t understand why the scattering presented in Figures 6-8 suggests that the IEO algorithm has the highest efficiency for the experimental I-V data. Highest efficiency in comparison with what?
3. What is the purpose of Figure 12?

Minor points
1. Page 2, line 51: Change “depend” to “depends”.
2. Page 6, line 7: Change “consequently” to “consecutively”.
3. The quality of Figure 4 should be improved.
4. Page 8, line 179: Change “to” to “is”.
5. Page 9, line 198: Change “best error” to “smaller errors”.
6. Page 9: Move the “are listed in Table 4” from after Table 3 to before Table 3.
7. Page 11, line 235: Change “solution” to “solutions”.
8. Page 11, last line: Perhaps change “I-V characteristics” to “I-V plots”.
9. Page 13, line 256: What is the purpose of the sentence starting with “In addition...”? In addition to what?
10. In Tables 11 and 12, add references to the described work.
11. In Figure 11, the labels of the y-axis are truncated.
12. Table 12 is stacked in the middle of a paragraph.
13. Page 15, line 283: Change “are accurate” to “are more accurate”.

---

## Round 0.3 · accepted · Accept

Date: 16/08/2021

Your manuscript entitled "An Efficient Equilibrium Optimizer for Parameters Identification of Photovoltaic Modules" has been accepted for publication in PeerJ Computer Science in its current form.

Reviewer 1 ·

Basic reporting

Very good

Experimental design

Very good

Validity of the findings

The findings are valid

Additional comments

The authors have satisfactory addressed all my comments and I now find the manuscript ready for publication.